# ROYAL SOCIETY
# OPEN SCIENCE

cognition/psychology

facial attractiveness, data-driven mathematical model, facial shape, facial reflectance

**Author for correspondence:**
Koyo Nakamura
e-mail: koyo@fennel.sci.waseda.ac.jp

# Data-driven mathematical model of East-Asian facial attractiveness: the relative contributions of shape and reflectance to attractiveness judgements

Koyo Nakamura[1,2,3] and Katsumi Watanabe[1,4]

[1]Faculty of Science and Engineering, Waseda University, Tokyo, Japan
[2]Japan Society for the Promotion of Science, Tokyo, Japan
[3]Keio Advanced Research Centers, Tokyo, Japan
[4]Art and Design, University of New South Wales, Sydney, Australia

KN, 0000-0002-9506-1644

Facial attractiveness is judged through a combination of multiple cues including morphology (facial shape) and skin properties (facial reflectance). While several studies have examined the way in which people in Western cultures judge facial attractiveness, there have been fewer investigations into non-Western attitudes. This is because stimuli that quantitatively vary the attractiveness of non-Western faces are rare. In the present study, we built a model of the attractiveness of East-Asian faces, judged by East-Asian observers. Therefore, 400 computer-generated East-Asian faces were created and attractiveness rating scores were collected from Japanese observers. Data-driven mathematical calculations were used to identify quantitative links between facial attractiveness and shape and reflectance properties, with no prior hypothesis. Results indicate that faces with larger eyes, smaller noses and brighter skin are judged as more attractive, regardless of the sex of the faces, possibly reflecting a general preference for femininity. Shape is shown to be a strong determinant of attractiveness for both male and female faces, while reflectance properties are less important in judging male facial attractiveness. Our model provides a tool to effectively produce East-Asian face stimuli that quantitatively varies attractiveness and can be used to elucidate visual processes related to attractiveness judgements.

# 1. Background

Facial attractiveness plays a pivotal role in social perception and has considerable impact on a wide range of social interactions. Attractive people tend to have more sexual partners [1] and earn a higher income [2]. Attractive faces elicit favourable treatment even in the cases of juror decisions [3], upbringing [4], scientific communication [5] and personality judgements [6]. Thus, being attractive results in beneficial social outcomes in real-world situations; humans show a strong positive bias toward attractive faces when making social decisions.

Given the pervasive influence of facial attractiveness, it is natural for psychologists to try to determine the features that make a face attractive [7]. Previous studies have identified multiple facial cues related to attractiveness judgements [8,9]. In general, such cues are identified in terms of facial morphology (facial shape) and skin properties (facial reflectance). For facial shape cues, averageness, symmetry and sexual dimorphism (masculinity for men and femininity for women) are well documented as influential determinants of facial attractiveness [7,10,11]. More specifically, faces with an average-looking shape, size and configuration [12] or a symmetric shape [13] are perceived as being more attractive than faces with a distinctive or asymmetric shape. Furthermore, shape differences between the sexes that emerge at puberty (i.e. sexual dimorphism) are related to attractiveness judgements; sex-typical facial characteristics are often associated with attractiveness [14,15]. Across cultures there is a general consensus that female-looking female faces with larger eyes and pronounced cheekbones are preferred to male-looking female faces, irrespective of the sex of the evaluator [16,17]. However, a preference for the masculinity of male faces, with features such as larger jawbones and more prominent brow ridges, is not consistent [16,18,19]. Facial reflectance cues such as texture, colour, and contrast also affect attractiveness judgements, independently of facial shape cues [20–23]. For example, exaggerating yellowness, redness or lightness on a face increases perceived attractiveness [20,24]. The above-mentioned shape and reflectance characteristics may perhaps be preferred because they act as reliable predictors for potential physical health or fecundity, possibly leading to higher rates of reproductive success [15,25]. Consistent with the idea that facial attractiveness signals heritable fitness and physical health, the preferences for these facial features are biologically based and thus observed across Western and non-Western cultures [14,17,26].

In order to examine the contributions of multiple facial cues to attractiveness judgement, previous studies typically tested the effect of a specific and predetermined feature in a hypothesis-driven approach [27]. Researchers manipulated the target feature (e.g. eye size) parametrically, while the other features remained constant [12,13]. This approach has been used to isolate the effect of a targeted facial feature on attractiveness, and provided significant insights into the mechanisms of facial attractiveness [19]. Nevertheless, such a hypothesis-driven approach tends to overlook other potential cues that researchers have not taken into account in advance [27,28]. Given that faces vary along multiple dimensions, and that several of these variations simultaneously affect facial attractiveness judgement [8,29], another approach to capture facial variations is required in order to further elucidate the factors that make a face attractive.

Data-driven methods that may be used to complement the hypothesis-driven approach have increasingly attracted the attention of researchers [30–32]. These methods rely on building a model to represent the way in which faces vary on facial impressions (e.g. attractiveness, trustworthiness and dominance) by sampling multiple features from a variety of faces. Quantitative associations are then calculated between a specific impression and multidimensional facial features (e.g. shape and reflectance components) with no prior hypothesis. Oosterhof & Todorov [31] built a mathematical model describing social attributes related to faces (e.g. trustworthiness and dominance) using all possible face shape information from various computer-generated Caucasian faces. Models for facial trustworthiness and facial dominance were built, and so any arbitrary face could be varied along the trait dimensions by parametrically transforming the facial shapes. By following this approach, Oosterhof & Todorov [31] found that trustworthy faces are characterized by smiles while dominant faces are characterized by anger. Said & Todorov [8] also applied the data-driven method to the attractiveness of Caucasian faces and showed that data-driven modelling provides more precise predictions for attractiveness judgement than predictive models by using a combination of well-documented influential factors for facial attractiveness (i.e. averageness and sexual dimorphism). The data-driven modelling of facial attractiveness thus has great potential for discovering as yet unidentified facial cues to attractiveness in an unbiased and controlled manner, without any *a priori* constraints on attractiveness cues [8,27,29].

Whereas the hypothesis-driven studies have revealed facial features that contribute to attractiveness in different cultures including Western and non-Western countries [14,26,33], the data-driven approach has been used to examine faces exclusively in Western cultures, with the exception of one recent work [34]. Cross-cultural studies have shown that the criteria for facial attractiveness judgement vary across different cultures [17,24,35]. Therefore, we sought to examine variations in non-Western faces that affect attractiveness judgements in non-Western observers using the data-driven approach. In the present study, data-driven mathematical modelling of the facial attractiveness of East-Asian faces was conducted using computer-generated faces (FaceGen Modeller, Singular Inversions). In the FaceGen Modeller, individual faces are represented by a combination of facial shape and face reflectance components. Facial shape corresponds to the combination of positions and shapes of facial features (e.g. eyes and chin), which are represented by the vertex positions of a polygonal model of fixed mesh topology [31]. Face reflectance includes facial properties such as brightness, colour and texture variations on the surface map of the faces [27,36]. Computer-generated faces allow greater control over facial features, and thus can be used for identifying the associations between facial features and attractiveness with greater precision and effectiveness when compared to real faces [27].

In this study, we first sampled many different faces and collected ratings on their attractiveness, then built a data-driven model of facial attractiveness for both male and female faces. We then investigated the relative contributions of facial shape and facial reflectance information to attractiveness judgement. The relative importance of facial shape and reflectance on attractiveness judgement is still under debate [37–39]. One study using computer-generated Caucasian faces showed that shape information is more important than reflectance information in the case of both male and female attractiveness judgements [39]. However, another study using photographed Caucasian faces found that both shape and reflectance components are equally important for female attractiveness, whereas the reflectance component is more important than shape in the case of male attractiveness [37]. The use of computer-generated faces in this study enabled us to manipulate facial shape and facial reflectance independently in order to examine the relative contributions of facial shape and facial reflectance on attractiveness judgements in Japanese observers.

# 2. Method

## 2.1. Participants

A model training study involved the participation of 10 Japanese males and 10 Japanese females (mean age = 19.35, s.d. = 1.31). A separate group of Japanese 24 males and 24 females were recruited for a model validation study (mean age = 20.33, s.d. = 1.71). All participants had normal or corrected-to-normal vision and were naive to the purpose of the study. The study was approved by the institutional review board (IRB) of Waseda University (2015–033). All procedures were carried out in accordance with the Declaration of Helsinki. Written informed consent was obtained from all participants in advance.

## 2.2. Apparatus and stimuli

All the face stimuli were created using FaceGen Modeller (Singular Inversions, Toronto, Canada). We generated East-Asian male and female faces by setting the FaceGen's race control to East-Asian and by setting the gender control from −4 (extremely masculine) to +4 (extremely feminine). Male faces were randomly generated with a gender control value between −4 and −1 and female faces were generated with a value between +1 and +4. All faces were viewed front-on and were emotionally neutral. The age was set to 20 years old for all the faces.

All the faces were generated using the face space model implemented in FaceGen, in which faces are represented using 100 dimensions (50 shape dimensions and 50 reflectance dimensions; for more details, see Procedure). In order to obtain a diverse set of faces, we first generated 10 000 faces for each sex drawn from random combinations of values for the 100 principal components (PCs), and then selected 200 faces that provided the greatest differences, one from the others, on the basis of the average Euclidean distance to all the other faces [40]. After the face generation, we scaled the values of the 100 PCs by a factor of 0.5, rendering their appearance close to the average face. This scaling maintains the relative differences between the faces, while it reduces facial distinctiveness. For a model validation study, we randomly generated another set of 10 male and 10 female faces that were not used in a model training study.

**Table 1.** Descriptive statistics of attractiveness rating score.

| | male faces | | | female faces | | |
|---|---|---|---|---|---|---|
| | mean | s.d. | range | mean | s.d. | range |
| male participants | 4.56 | 0.98 | 1.90 – 7.20 | 4.67 | 0.99 | 2.40 – 7.60 |
| female participants | 4.33 | 1.06 | 2.20 – 7.20 | 4.59 | 0.94 | 2.50 – 7.20 |

The experiment was performed using Matlab running on a MacMini. All the stimuli were presented on a 23.5 inch LED monitor (FORIS FG2421, EIZO) with a refresh rate of 100 Hz and a screen resolution of 1920 × 1080 pixels. Participants viewed the monitor from a viewing distance of 57 cm.

## 2.3. Procedure

### 2.3.1. Model training

Each participant sat in front of the computer monitor and was given the explanation for the procedure of attractiveness rating task. In the attractiveness rating task, participants were asked to rate the attractiveness of 200 male and 200 female faces on a scale ranging from 1 (least attractive) to 9 (most attractive). In each trial, following the presentation of a fixation cross for 500 ms, a face was presented on the centre of the screen. Participants were able to view the face until their response was made and were told to rate each face with subjective but relative criteria. The participants completed two separate sessions, within each of which the sex of the faces was fixed. Within a session, the faces were presented in a random order. The order of the sessions (i.e. the sex of the faces) was counterbalanced across participants.

### 2.3.2. Data-driven mathematical modelling of facial attractiveness

In order to identify facial shape and facial reflectance information associated with attractiveness, we modelled an attractiveness dimension using a data-driven mathematical approach. In the model, the average facial shape and reflectance are represented according to the following expressions:

$$\bar{S} = (\bar{x}_1, \bar{y}_1, \bar{z}_1, \ldots, \bar{x}_{N_s}, \bar{y}_{N_s}, \bar{z}_{N_s}) \in \mathfrak{R}^{3N_s}$$
$$\text{and} \quad \bar{T} = (\bar{r}_1, \bar{g}_1, \bar{b}_1, \ldots, \bar{r}_{N_t}, \bar{g}_{N_t}, \bar{b}_{N_t}) \in \mathfrak{R}^{3N_t}$$

where $N_s = 2043$ and $N_t = 256 \times 256$ [41]. In the FaceGen Modeller, faces can be represented in a lower-dimensional face space through principal component analysis (PCA). PCA is implemented in order to reduce dimensionality, and thus we used 50 dimensions to represent face shape and 50 dimensions to represent facial reflectance. The extent to which facial shape is altered as the $i$th PC is changed, 1 s.d., is represented by $\Delta_i$ according to the expression

$$\Delta_i = (\Delta x_{1i}, \Delta y_{1i}, \Delta z_{1i}, \ldots, \Delta x_{ni}, \Delta y_{ni}, \Delta z_{ni}) \in \mathfrak{R}^{3n}.$$

Thus, a face $S$ is represented as an average face $\bar{S}$ plus a weighted sum of the shape eigenfaces, as given by

$$S = \bar{S} + \sum_{i=1}^{50} \alpha_i \cdot \Delta_{i\bullet},$$

where the shape vector $\alpha \in \mathfrak{R}^{3n}$ maps the PC offset vectors $\Delta_{i\bullet}$ onto the original geometry of the face. The same logic was applied to the reflectance dimensions, and 50 PCs for reflectance dimensions were treated in a similar manner.

Next, facial attractiveness was modelled as linear combinations of 100 PCs, more precisely, as the best linear fit of the mean attractiveness rating score $r \in \mathfrak{R}^{200}$ as a function of the 100 PCs. Descriptive statistics of the attractiveness rating scores are summarized in table 1. Given that the inter-rater reliability (Cronbach's alpha) of the attractiveness rating was so high for both male ($\alpha = 0.93$) and female faces ($\alpha = 0.95$), we used mean attractiveness rating scores across all the participants in the following analysis. Here the attractiveness rating scores were standardized for each participant.

Modelling of facial attractiveness was implemented separately for male and female faces. The optimal direction for the attractiveness vector is calculated using the expression

$$t = F \cdot r, \quad \hat{t} = \frac{t}{\|t\|},$$

where $F \in \Re^{50 \times 200}$ for shape dimensions. The attractiveness of any arbitrary face was changed $\delta$ s.d. according to

$$\alpha' = \alpha + \delta \cdot \hat{t}.$$

### 2.3.3. Model validation

In order to validate our model of facial attractiveness and evaluate the relative contributions of facial shape and facial reflectance to attractiveness judgements, we applied three types of attractiveness manipulation to 20 novel randomly generated faces: shape-and-reflectance, shape-only and reflectance-only manipulation. For each manipulation, we created seven versions of the 20 faces, varying the attractiveness level from $-3$ (less attractive) to $+3$ (more attractive) in s.d. units. The shape-only manipulation involved a change in the facial shape components while keeping the reflectance components constant (0 s.d.), and the reflectance-only manipulation involved changes to the facial reflectance components while keeping the shape components constant. The shape-and-reflectance manipulation involved changes to both the shape and reflectance components. Participants were randomly assigned to one of the three manipulation conditions and were asked to rate the facial attractiveness of a set of male and female faces.

### 2.3.4. Statistical analysis

In order to ascertain that our model-based manipulations reliably predict perceived facial attractiveness, we regressed mean attractiveness rating scores across participants onto face exaggeration ($-3$, $-2$, $-1$, 0, $+1$, $+2$, $+3$ s.d. on attractiveness dimension) with Bayesian linear regression models. The regression analysis was carried out separately for the three types of face manipulation for male and female faces. Furthermore, to better evaluate the relative contributions of shape and reflectance information to facial attractiveness judgements, we performed a Bayesian estimation of Pearson's correlation coefficients between the attractiveness ratings. The correlation coefficient between shape-only and shape-and-reflectance conditions, and the correlation coefficient between reflectance-only and shape-and-reflectance conditions were estimated and compared.

All the analyses were performed in R (version 3.5.1) using the 'rstan' package. All iterations were set to 5000, and the burn-in samples were set to 500, with the number of chains set to 4. The value of *Rhat* for all parameters equalled 1.0, indicating convergence across the four chains. The expected *a posteriori* (EAP) and 95% credible interval (CrI) were used as representative values for the estimated parameters.

## 3. Results

Figure 1 shows the variation of facial shape and reflectance components with our model-based attractiveness manipulation, with five versions of an arbitrary male and female face that differ in attractiveness ($-10$ s.d., $-3$ s.d., 0 s.d., $+3$ s.d., $+10$ s.d.).

### 3.1. Shape-and-reflectance manipulation

In order to determine that the data-driven manipulation of facial shape and reflectance components successfully predicts perceived attractiveness, we regressed the mean attractiveness rating scores across the participants onto facial manipulations ($-3$, $-2$, $-1$, 0, $+1$, $+2$, $+3$ s.d. on the attractiveness dimension). The result showed that the attractiveness rating score significantly changed with model-based shape and reflectance manipulation (figure 2a) for both male faces ($\beta = 0.59$, 95%CrI = [0.51−0.68]) and female faces ($\beta = 0.78$, 95%CrI = [0.71−0.86]).

(*a*) shape and reflectance

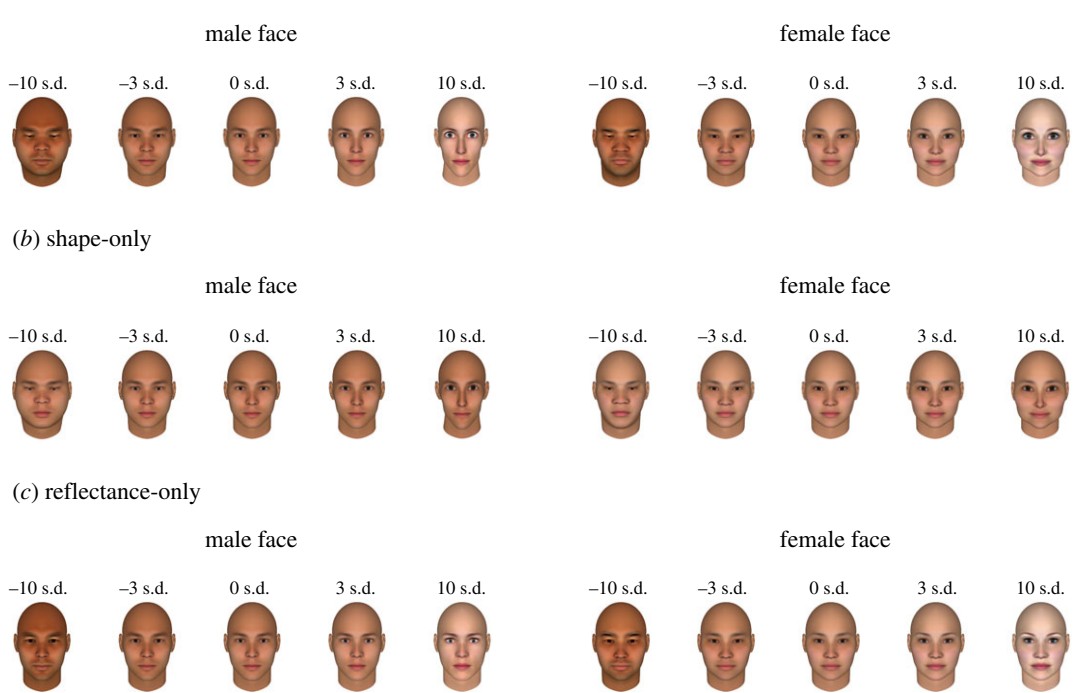

**Figure 1.** Example of manipulated faces using the data-driven mathematical model for facial attractiveness. Two faces, one male and one female, are transformed in terms of both shape and reflectance dimensions (*a*), only the shape dimension (*b*), and only the reflectance dimension (*c*). The extent of the facial manipulation is presented in s.d. units.

## 3.2. Shape-only manipulation

In a similar vein, we tested the effectiveness of facial shape modulation using a linear regression model. The attractiveness rating score varied with shape manipulation (figure 2*b*) for both male faces ($\beta = 0.58$, 95%CrI = [0.49–0.67]) and female faces ($\beta = 0.45$, 95%CrI = [0.36–0.54]).

## 3.3. Reflectance-only manipulation

Similarly, we also tested the effectiveness of facial reflectance modulation using a linear regression model. For the male faces, the attractiveness rating did not vary with face reflectance manipulation ($\beta = 0.04$, 95%CrI = [−0.07–0.16]) while in the case of the female faces, the attractiveness rating changed with face reflectance manipulation ($\beta = 0.36$, 95%CrI = [0.27–0.46]; figure 2*c*).

## 3.4. Relative contributions of facial shape and reflectance

In order to evaluate the relative contributions of facial shape and facial reflectance information in our data-driven model of facial attractiveness, we estimated the correlation coefficients between ratings for shape-and-reflectance manipulated faces and shape-manipulated faces, and between ratings for shape-and-reflectance manipulated faces and reflectance-manipulated faces. This was performed separately for the male and female faces using Bayesian estimation (figure 3). For male faces, ratings of shape-and-reflectance manipulated faces were positively correlated with both the ratings of shape-manipulated faces ($\rho = 0.93$, 95%CrI = [0.90–0.96]) and reflectance-manipulated faces ($\rho = 0.41$, 95%CrI = [0.20–0.59]). Furthermore, we found that the rating of shape-manipulated faces showed a stronger correlation with that of shape-and-reflectance manipulated faces when compared to the rating of reflectance-manipulated faces was ($\Delta\rho = 0.52$, 95%CrI = [0.34–0.74]). For female faces, the ratings of shape-and-reflectance manipulated faces were positively correlated with both the ratings of shape-manipulated faces ($\rho = 0.86$, 95%CrI = [0.78–0.91]) and reflectance-manipulated faces ($\rho = 0.81$, 95%CrI = [0.72–0.88]). In addition, there were no statistically significant difference between their corresponding correlation coefficients ($\Delta\rho = 0.05$, 95%CrI = [−0.06–0.15]).

**Figure 2.** Validation of the data-driven model for East-Asian facial attractiveness by Bayesian linear regression. Plotted data points indicate the mean attractiveness rating scores and shaded areas represent 95% credible intervals. Attractiveness ratings were given to faces that were manipulated in terms of both shape and reflectance dimensions (*a*), only shape dimensions (*b*), and only reflectance dimensions (*c*). The extent of facial manipulation is presented in s.d. units.

## 4. Discussion

This study aimed to build a model determining the perceived attractiveness of East-Asian faces using a data-driven approach. Our study is the first to model the facial attractiveness of computer-generated East-Asian faces based on the attractiveness ratings of Japanese observers using a data-driven method, with the majority of previous studies focusing on the social perception of Caucasian faces [30,31]. Understanding social perceptions, including those related to attractiveness, of non-Western faces is key to examining the universality and diversity of social judgements that are based on faces. In contrast to the hypothesis-driven approach, in which the effect of a predetermined facial component is individually tested, we employed a data-driven approach in order to determine the way in which facial shape and reflectance components covary with attractiveness. Our model enables the

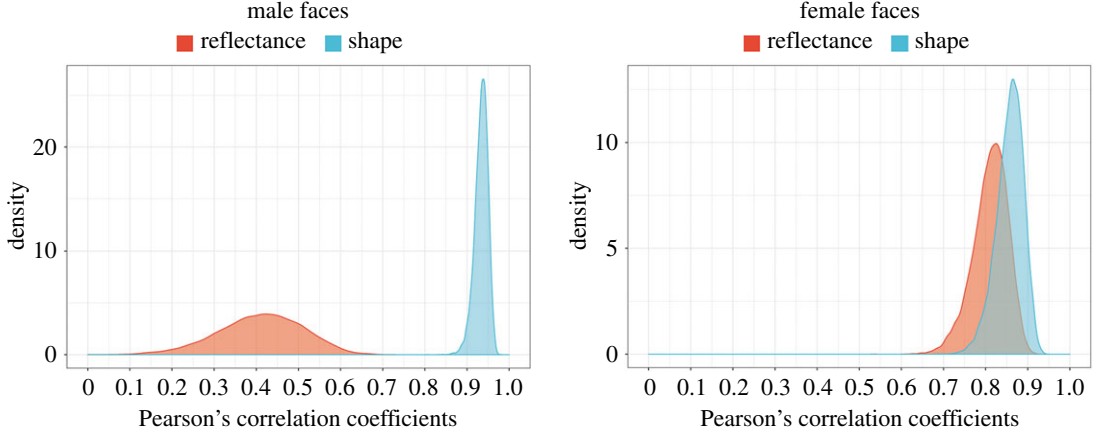

**Figure 3.** The relative contributions of facial shape and reflectance information on attractiveness judgements. Pearson's correlation coefficients between ratings of shape-and-reflectance manipulated and shape-manipulated faces, and the correlation coefficients between ratings of shape-and-reflectance manipulated and reflectance-manipulated faces were estimated separately for male and female faces.

quantitative manipulation of the attractiveness of any arbitrary East-Asian faces by transforming facial shape and reflectance components.

By visualizing the changes in facial features along the attractiveness dimension, we confirmed that the facial attractiveness of East-Asian faces is characterized by both facial shape and reflectance characteristics. Although both global configuration and local parts of faces covary with perceived attractiveness, it is of note that bigger eyes, smaller noses and brighter skin in both male and female faces lead to an increase in perceived attractiveness[1] (figure 1). According to Cunningham [42], faces with bigger eyes and a smaller nose are perceived to be younger and more immature. Therefore, such features are preferred by both male and female observers [14,17]. In particular, a preference for larger eyes has often been observed when judging female facial attractiveness [43]. Such facial characteristics can be related to feminine traits that are formed under the influence of oestrogen [44], where evolutionary psychology posits that the oestrogen-related traits of female faces can be signals for reproductive value, indicative of youth and high fecundity [7,15]. Thus, male preferences for feminine female faces can reflect an adaptation for choosing high-quality mates, yielding advantages in terms of reproduction [15]. Notably, a recent study has shown that amongst members of various Western and Asian countries, Japanese males show a particular preference for feminine female faces, although the reason for this prominent preference for femininity in Japanese men has not yet been identified [35].

Our findings are also consistent with previous hypothesis-driven studies showing that feminine-looking male and female faces are preferred over masculine-looking faces across cultures [14]. Whereas the masculinity of male faces signals a high genetic quality in terms of potential mating and health [25], it can also be associated with negative personality traits and behaviours. For example, masculine male faces are perceived as less emotionally warm, less honest and less cooperative than feminine male faces [14]. Moreover, it has been reported that highly masculine men are more likely to be subject to marital problems and divorce when compared to more feminine men [45], and that masculine men are insensitive to infant cries, feeling less sympathetic than feminine men [46]. Preferences for a feminized male face might be derived from an avoidance of such a man as a long-term partner and may reflect desired personality traits [47]. Together with previous findings, the current results indicated that Japanese people prefer femininity when judging the facial attractiveness of both male and female faces.

One key point is that the relative contributions of facial shape and reflectance information to attractiveness judgements differ between male and female East-Asian faces. For male faces, the linear transformation of facial shape components along the attractiveness dimension while reflectance components remain fixed significantly changed perceived attractiveness, while the linear transformation of facial reflectance components while shape components were fixed did not affect perceived

---

[1]Note that the critical facial cues to attractiveness were identified by visualizing face exaggeration of facial features along the attractiveness dimensions (see the faces with −10 s.d. and +10 s.d. in figure 1).

attractiveness. As such, facial attractiveness appears to be mainly determined by the facial shape information. For female faces, both facial shape and reflectance transformations significantly altered perceived attractiveness, and the contributions of facial shape and reflectance information to female face attractiveness judgements were equally high. Our results agree with those of previous studies, showing high contributions of both facial shape and reflectance information on female attractiveness judgement [37]. However, our findings with respect to male facial attractiveness are partially consistent with previous findings related to Caucasian faces [37–39]. For male attractiveness, one study reported that shape information was more important than reflectance information [39], while the other suggested that reflectance information was more important than shape information [38]. No unified explanation for such inconsistencies has yet been proposed. With regards to this point, recent studies indicated that the femininity of male faces contributes to attractiveness in terms of facial shape, and masculinity contributes to attractiveness in terms of facial reflectance, where a darker and redder skin is preferred [8,38]. Given that Japanese participants tend to show the strongest preference for femininity of male faces among those in both Western and non-Western cultures [35], the preferences for femininity in the male facial shape and masculinity in the male facial reflectance might be stronger and weaker, respectively, than assumed, leading to a greater contribution from the facial shape information. Given that male attractiveness judgement may be more complicated and condition-dependent than expected, further research is needed in order to replicate our results and investigate the factors that determine the relative contributions of facial shape and reflectance information to attractiveness judgements.

The present study has been subject to associated limitations that are important to consider. Firstly, we used computer-generated faces because this allowed for standardization in terms of facial features, lighting conditions and extra-facial information (e.g. head hair), thereby enabling a more precise identification of facial cues critical to attractiveness judgements. However, the computer-generated faces might not fully capture the real-life rich cues present in human faces. In particular, the computer-generated faces appear to lack skin textures that appear on real faces, such as wrinkles and blemishes [48]. Given the evidence that skin smoothness and blemishes affect perceived attractiveness [23], future studies would need to incorporate such features in the data-driven model. Secondly, although our model is based on the aggregated rating scores across the participants because inter-rater reliability was sufficiently high, there are often individual differences in attractiveness judgements [49,50]. It is possible that reflectance information contributes to attractiveness judgements less than shape information because the preference for reflectance features varies across participants, not simply because the reflectance information is less important in judging facial attractiveness. This possible factor should be tested in future studies by modelling shared and private tastes in facial attractiveness in a data-driven manner.

## 5. Conclusion

Quantitative links between the facial attractiveness and shape and reflectance information of East-Asian faces were determined on the basis of data-driven mathematical calculations. The values for data-driven modelling of attractiveness were then verified using non-Western faces. Whereas model-based facial shape transformation was sufficient to increase the attractiveness of male faces, both shape and reflectance transformations were effective to increase the attractiveness of female faces. Our model converged well with the previous finding of the hypothesis-driven studies, and provides a tool that can effectively produce East-Asian face stimuli with quantitatively variable attractiveness. Our data-driven study demonstrated that attractiveness of East-Asian faces is judged through a combination of multiple cues, and isolating the effect of each cue by the hypothesis-driven approach would benefit further understanding of visual processes related to attractiveness judgements.

Ethics. The study was approved by the institutional review board (IRB) of Waseda University (2015–033). All procedures were carried out in accordance with the Declaration of Helsinki. Written informed consent was obtained from all participants in advance.

Data accessibility. Raw data were available from the Dryad Digital Repository (https://datadryad.org/resource/doi:10. 5061/dryad.6vm4qs4) [51].

Authors' contributions. K.N. designed the study, collected and analysed the data, and wrote the initial draft of the manuscript. K.W. designed the experiment and critically reviewed and co-wrote the manuscript.

Competing interests. We declare we have no competing interests.

Funding. This work was supported by Grant-in-Aid for JSPS Research Fellow to K.N. (17J04125) and Grant-in-Aid for Scientific Research on Innovative Areas (17H06344) and Strategic Japanese-Swiss Science and Technology Programme from JSPS to K.W.

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
