## [Reviewer comments · Royal Society Open Science]

Review History

RSOS-182189.R0 (Original submission)

Review form: Reviewer 1 (Alex Todorov)

Is the manuscript scientifically sound in its present form?

Yes

Are the interpretations and conclusions justified by the results?

Yes

Is the language acceptable?

Yes

Is it clear how to access all supporting data?

No

Do you have any ethical concerns with this paper?

No

Have you any concerns about statistical analyses in this paper?

No

Recommendation?

Accept as is

Comments to the Author(s)

This is a well-written and clear manuscript applying data-driven computational methods to reveal the facial features that correlate with perceptions of attractiveness. While these methods have been used to model various social perceptions, they have never been applied to East Asian faces and East Asian observers. Given cross-cultural differences in perception and cognition, this is an important contribution. It further showcases the utility and discovery value of data-driven methods.

The work was solid and the results straightforward. It was surprising to find out that for male faces, the texture properties making them more attractive are lighter; this is the opposite finding for Caucasian male faces in Said and Todorov. But this is part of the value of data-driven methods. This cross-cultural difference is worth exploring in the future.

There is another recent paper (on the open science framework) that uses the same approach. It would be good to acknowledge this work (<https://osf.io/jurcq/>). As in Said and Todorov, this group worked with Caucasian faces, but real (female) faces. The three papers taken together demonstrate the value of data-driven approaches.

Review form: Reviewer 2

Is the manuscript scientifically sound in its present form?

Yes

Are the interpretations and conclusions justified by the results?

Yes

Is the language acceptable?

Yes

Is it clear how to access all supporting data?

No

Do you have any ethical concerns with this paper?

No

Have you any concerns about statistical analyses in this paper?

Yes

Recommendation?

Major revision is needed (please make suggestions in comments)

Comments to the Author(s)

This study attempted to fill a gap in the facial attractiveness literature through its use of a large set of images of computer-generated male and female East Asian faces and through attractiveness

judgments of these faces by East Asian perceivers. The study was exploratory, data-driven (i.e., bottom-up), and sought to determine the extent to which facial shape and facial surface contribute to the attractiveness judgments.

The authors examined the relative contributions of facial shape and facial surface on attractiveness judgments of faces. These two factors need to be discussed in greater depth as it is unclear what each exactly signifies. For example, does facial surface refer to the darkness and color characteristics of the skins of the faces? This seems to be the case when looking at Figure 1. If so, why don't the authors just call it darkness and color? The word "surface" is misleading because it denotes texture, blemishes, etc. which the authors acknowledge not including as factors in the study.

The participants rated the faces for attractiveness on a 9-point scale. It would be helpful for the readers if some descriptive statistics regarding these ratings are provided. Figure 2 shows some of these data, but the raw values could help especially when considering the present findings in light of those of previous studies.

One of the main findings of the study is "that bigger eyes, smaller noses, and brighter skin in both male and female faces lead to an increase in perceived attractiveness." What specific analysis led to this conclusion? Is it based merely on an examination of the faces presented in Figure 1, or are there quantitative values to support this? Perhaps I missed something but this is not clear.

Finally, I would like to push the authors' approach a bit further. The data-driven approach showed some interesting findings. However, would it not make scientific sense to also have included a hypothesis-driven component that used the East-Asian faces to test whether classic facial attractiveness effects would obtain for these faces? If the issue is that only Caucasian faces have been used, then the first step should have been to use the East-Asian faces to try to replicate previous studies. Then extend this by using the data-driven approach.

Decision letter (RSOS-182189.R0)

27-Feb-2019

Dear Dr Nakamura,

The editors assigned to your paper ("Data-driven mathematical model of East-Asian facial attractiveness:

The relative contributions of facial shape and surface to attractiveness judgments") have now received comments from reviewers. We would like you to revise your paper in accordance with the referee and Associate Editor suggestions which can be found below (not including confidential reports to the Editor). Please note this decision does not guarantee eventual acceptance.

Please submit a copy of your revised paper before 22-Mar-2019. Please note that the revision deadline will expire at 00.00am on this date. If we do not hear from you within this time then it will be assumed that the paper has been withdrawn. In exceptional circumstances, extensions may be possible if agreed with the Editorial Office in advance. We do not allow multiple rounds of revision so we urge you to make every effort to fully address all of the comments at this stage. If deemed necessary by the Editors, your manuscript will be sent back to one or more of the

original reviewers for assessment. If the original reviewers are not available, we may invite new reviewers.

- Data accessibility

If you wish to submit your supporting data or code to Dryad (<http://datadryad.org/>), or modify your current submission to dryad, please use the following link:
<http://datadryad.org/submit?journalID=RSOS&manu=RSOS-182189>

- Competing interests

- Authors' contributions

- Acknowledgements

- Funding statement

on behalf of Professor Essi Viding (Subject Editor)
openscience@royalsociety.org

Associate Editor's comments:

Please ensure you fully respond to the reviewers' concerns (if particular, those articulated by Referee 2).

Comments to Author:

Reviewers' Comments to Author:

Reviewer: 1

Comments to the Author(s)

This is a well-written and clear manuscript applying data-driven computational methods to reveal the facial features that correlate with perceptions of attractiveness. While these methods have been used to model various social perceptions, they have never been applied to East Asian faces and East Asian observers. Given cross-cultural differences in perception and cognition, this is an important contribution. It further showcases the utility and discovery value of data-driven methods.

The work was solid and the results straightforward. It was surprising to find out that for male faces, the texture properties making them more attractive are lighter; this is the opposite finding for Caucasian male faces in Said and Todorov. But this is part of the value of data-driven methods. This cross-cultural difference is worth exploring in the future.

There is another recent paper (on the open science framework) that uses the same approach. It would be good to acknowledge this work (<https://osf.io/jurcq/>). As in Said and Todorov, this

group worked with Caucasian faces, but real (female) faces. The three papers taken together demonstrate the value of data-driven approaches.

Reviewer: 2

Comments to the Author(s)

This study attempted to fill a gap in the facial attractiveness literature through its use of a large set of images of computer-generated male and female East Asian faces and through attractiveness judgments of these faces by East Asian perceivers. The study was exploratory, data-driven (i.e., bottom-up), and sought to determine the extent to which facial shape and facial surface contribute to the attractiveness judgments.

The authors examined the relative contributions of facial shape and facial surface on attractiveness judgments of faces. These two factors need to be discussed in greater depth as it is unclear what each exactly signifies. For example, does facial surface refer to the darkness and color characteristics of the skins of the faces? This seems to be the case when looking at Figure 1. If so, why don't the authors just call it darkness and color? The word "surface" is misleading because it denotes texture, blemishes, etc. which the authors acknowledge not including as factors in the study.

The participants rated the faces for attractiveness on a 9-point scale. It would be helpful for the readers if some descriptive statistics regarding these ratings are provided. Figure 2 shows some of these data, but the raw values could help especially when considering the present findings in light of those of previous studies.

One of the main findings of the study is "that bigger eyes, smaller noses, and brighter skin in both male and female faces lead to an increase in perceived attractiveness." What specific analysis led to this conclusion? Is it based merely on an examination of the faces presented in Figure 1, or are there quantitative values to support this? Perhaps I missed something but this is not clear.

Finally, I would like to push the authors' approach a bit further. The data-driven approach showed some interesting findings. However, would it not make scientific sense to also have included a hypothesis-driven component that used the East-Asian faces to test whether classic facial attractiveness effects would obtain for these faces? If the issue is that only Caucasian faces have been used, then the first step should have been to use the East-Asian faces to try to replicate previous studies. Then extend this by using the data-driven approach.

Author's Response to Decision Letter for (RSOS-182189.R0)

See Appendix A.

RSOS-182189.R1 (Revision)

Review form: Reviewer 2

Is the manuscript scientifically sound in its present form?

Yes

Are the interpretations and conclusions justified by the results?

Yes

Is the language acceptable?

Yes

Is it clear how to access all supporting data?

Not Applicable

Do you have any ethical concerns with this paper?

No

Have you any concerns about statistical analyses in this paper?

No

Recommendation?

Accept as is

Comments to the Author(s)

The revised version of the manuscript is an improvement from the initial version. The use of "facial reflectance" is less misleading and more appropriately reflects what the authors examined. The authors also now make clear why they employed a data-driven instead of a hypothesis-driven approach. The table of attractiveness ratings data is a nice addition. Overall, I believe that the revision addressed the issues that were previously raised.

Decision letter (RSOS-182189.R1)

10-May-2019

Dear Dr Nakamura,

I am pleased to inform you that your manuscript entitled "Data-driven mathematical model of East-Asian facial attractiveness: The relative contributions of shape and reflectance to attractiveness judgements" is now accepted for publication in Royal Society Open Science.

on behalf of Prof Essi Viding (Subject Editor)
openscience@royalsociety.org

Associate Editor Comments to Author:

After further review, we're pleased to report the reviewer is satisfied by your manuscript's changes, and they now recommend publication - congratulations! Thank you for choosing Royal Society Open Science for your research, and we hope you'll submit further papers to the journal in future.

Reviewer comments to Author:

Reviewer: 2

Comments to the Author(s)

The revised version of the manuscript is an improvement from the initial version. The use of "facial reflectance" is less misleading and more appropriately reflects what the authors examined. The authors also now make clear why they employed a data-driven instead of a hypothesis-driven approach. The table of attractiveness ratings data is a nice addition. Overall, I believe that the revision addressed the issues that were previously raised.

Appendix A

March 19, 2019

Dear Editor at Royal Society Open Science:

We are grateful to have the opportunity to revise our manuscript entitled, “Data-driven mathematical model of East-Asian facial attractiveness: The relative contributions of facial shape and surface to attractiveness judgments” (Manuscript ID RSOS-182189), which has been reviewed for publication in *Royal Society Open Science*. Please find our revised manuscript, including three figures and one table. We revised our manuscript in accordance with two reviewers’ comments. First, we used the word “facial reflectance” instead of “surface” in our revised manuscript to appropriately describe our results that facial colour and brightness are important cues to attractiveness. Accordingly, we changed the title of our manuscript into “Data-driven mathematical model of East-Asian facial attractiveness: The relative contributions of shape and reflectance to attractiveness judgements” and provided a definition of facial shape and reflectance in *Introduction* in the revised manuscript. Second, we presented the faces transformed along the attractiveness dimension extremely (i.e., -10 *SD* and +10 *SD*) in Figure 1 to illustrate more clearly how facial features vary with attractiveness. These exaggerated faces were not used in the experiment, but it would be helpful for readers to visually appreciate facial cues to attractiveness. We would be glad if our revisions satisfy the reviewers’ essential suggestions.

Thank you for your consideration. I look forward to hearing from you.

Sincerely,

Koyo Nakamura

Faculty of Science and Engineering, Waseda University, 3-4-1, Ohkubo, Shinjuku, Tokyo, 169-8555, Japan.

Email: koyo@fennel.sci.waseda.ac.jp

Response to Referees

To Reviewer #1:

We appreciate your helpful comments on our manuscript. According to your suggestion, we checked the article in OSF (Holzleitner et al., 2018), and found that the study applied the data-driven approach to real Caucasian female faces in order to reveal important components in perception of facial attractiveness. We think this article deserves to be cited in our manuscript to discuss the value of the data-driven computational modeling (*lines 83-86, 103-106*).

To Reviewer #2:

We appreciate your helpful comments for important issues in this article. As we described below, we revised our manuscript according to your comments. We hope this revise meets your requests appropriately.

Reviewer's comment 1)

Reviewer #1 noted “The authors examined the relative contributions of facial shape and facial surface on attractiveness judgments of faces. These two factors need to be discussed in greater depth as it is unclear what each exactly signifies. For example, does facial surface refer to the darkness and color characteristics of the skins of the faces? This seems to be the case when looking at Figure 1. If so, why don't the authors just call it darkness and color? The word “surface” is misleading because it denotes texture, blemishes, etc. which the authors acknowledge not including as factors in the study.

Authors' response:

As Reviewer #2 pointed out, the word “facial surface” might be misleading for readers because the word conjures only skin properties such as “blemish” and “skin smoothness.” Therefore, we are convinced to use another word to signify multiple facial features such as brightness, colours, texture variation on the skin surface, which covary with attractiveness. In Todorov and his colleague's studies using face stimuli generated by the FaceGen, the word “reflectance” has been used instead of “surface.” It is defined as including facial properties such as “brightness,

color, and texture variations on the surface map of the faces” (Todorov and Oosterhof, 2011). Given the consistency between the previous data-driven studies and the current study, we replace “surface” with “reflectance” throughout our revised manuscript with a detailed definition of “reflectance” given by Todorov and Oosterhof (2011). Thus, we have added the statements for defining facial shape and reflectance in *Introduction* with necessary references (*lines 115 - 121*).

Reviewer’s comment 2)

Reviewer #2 noted “The participants rated the faces for attractiveness on a 9-point scale. It would be helpful for the readers if some descriptive statistics regarding these ratings are provided. Figure 2 shows some of these data, but the raw values could help especially when considering the present findings in light of those of previous studies. “

Authors’ response:

As Reviewer #2 suggested, it would be helpful to present descriptive statistics of facial attractiveness rating in order for readers to compare the present results with previous ones and to replicate our findings. In our revised manuscript, we reported mean score, standard deviation, range (min and max) of facial attractiveness ratings in Table 1 (*lines 207-208*).

Reviewer’s comment 3)

Reviewer #2 noted “One of the main findings of the study is “that bigger eyes, smaller noses, and brighter skin in both male and female faces lead to an increase in perceived attractiveness.” What specific analysis led to this conclusion? Is it based merely on an examination of the faces presented in Figure 1, or are there quantitative values to support this? Perhaps I missed something but this is not clear. “

Authors’ response:

In the data-driven approach, facial features driving attractiveness were identified through a mathematical calculation. More precisely, “*t*” (*line 214*) in our revised manuscript represents how facial features vary on the attractiveness dimension. For readers, however, the 100 dimensional vector “*t*” seems unmeaningful so that we visualized how facial features were changed when “*t*” was changed. In revised Figure 1, we presented the faces with extreme changed of “*t*” (-10SD and +10SD) to help readers visually appreciate which features vary along the attractiveness dimension. A similar technique was used in Oosterhof & Todorov (2008). Thus,

we also stated in a footnote that our conclusion is based on visualization of the attractiveness vector “ t ” (*line 542-544*).

Reviewer’s comment 4)

Reviewer #2 noted “Finally, I would like to push the authors’ approach a bit further. The data-driven approach showed some interesting findings. However, would it not make scientific sense to also have included a hypothesis-driven component that used the East-Asian faces to test whether classic facial attractiveness effects would obtain for these faces? If the issue is that only Caucasian faces have been used, then the first step should have been to use the East-Asian faces to try to replicate previous studies. Then extend this by using the data-driven approach.”

Authors’ response:

As Reviewer #2 commented, the first step to examine cultural differences in attractiveness judgements could be to take a hypothesis-driven approach to Western and non-Western cultures. As we stated in our revised manuscript, the hypothesis-driven studies have revealed that some facial cues to attractiveness such as the sexual dimorphism and symmetry affect attractiveness judgements universally across Western and non-Western cultures including Asian countries (*lines 72-75, 107-110*). Although the hypothesis-driven approach has emphasized the universality of perception of facial attractiveness and the idea was true to some extent, the data-driven approach has a potential to identify as yet unidentified facial cues that have not reported yet in the hypothesis-driven approach. This was the reason why we performed the data-driven computational modeling for East-Asian faces for the first time. To make clear this point, we stated the values and contributions of both hypothesis-driven and data-driven approaches in *Conclusion* section (*lines 383-393*).